# INTER-OBJECT COMMONSENSE RELATIONSHIP REASONING FOR SCENE GRAPH MANIPULATION

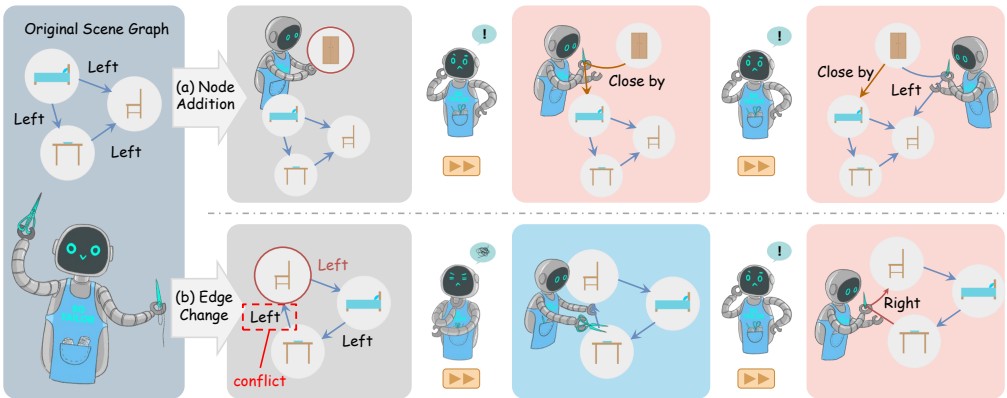

Figure 1: **SG-Tailor for scene graph manipulation.** SG-Tailor manipulates a given scene graph in two modes: (a) *Node Addition*, SG-Tailor autoregressively **reasons commonsense relationships** between a new node and existing nodes (*e.g.*, The wardrobe should be near the bed and to the left of the chair.). (b) *Edge Change*, it maintains the desired edge while **resolving conflicts** (*e.g.*, naively moving the chair to the left of the bed causes a conflict. SG-Tailor resolves this by replacing the conflicting edge to maintain coherence).

## ABSTRACT

Scene graphs capture complex relationships among objects, serving as strong priors for 3D scene generation and manipulation. However, reasonable manipulation of scene graphs remains a challenging and untouched task. The reasoning about a node's relationships with all other nodes is computationally intractable, as even a single edge modification can trigger conflicts due to the intricate relationship interdependencies within the graph. We introduce **SG-Tailor**, an autoregressive model that predicts the conflict-free relationships in a scene graph. SG-Tailor infers inter-object relationships, including generating commonsense edges for newly added nodes, and resolves conflicts arising from edge modifications to produce coherent, manipulated graphs for downstream tasks. When a new node is added, SG-Tailor queries the target node and forms node pairs with other nodes in the graph to predict the appropriate pairwise relationships. When an edge is modified, SG-Tailor employs a **Cut-And-Stitch** strategy to solve the conflicts and globally adjust the graph. Extensive experiments demonstrate that SG-Tailor outperforms competing methods by a large margin and can be seamlessly integrated as a plug-in module for scene generation and robotic manipulation tasks. The code will be released upon acceptance.

## 1 INTRODUCTION

Scene graphs effectively capture semantic relationships among objects by representing them as nodes and their interactions as edges (Li et al., 2024; Chang et al., 2021). This structured, interpretable representation is widely used in computer vision tasks, such as image captioning (Krishna et al., 2017), scene understanding (Zhang et al., 2021; Gu et al., 2024), and robotics applications (Hughes et al., 2022; Werby et al., 2024; Maggio et al., 2024; Jiang et al., 2024). Building on this foundation,

a highly flexible pipeline is envisioned for 3D data creation and manipulation using scene graphs. **(i)** extract a scene graph from visual data, **(ii)** manipulate it to enforce user-specified edits while preserving commonsense, and **(iii)** synthesize realistic outputs for downstream tasks. This streamlined approach significantly enhances flexibility and precision in interactive data generation conditioned on the large amount of 3D scene content (Jia et al., 2024; Wald et al., 2020; Ramakrishnan et al., 2021).

While extensive work has focused on extracting scene graphs from images or 3D data (Wu et al., 2021; Rosinol et al., 2021; Johnson et al., 2015; Im et al., 2024) and on synthesizing scenes from graphs (Johnson et al., 2018; Zhai et al., 2023; Yang et al., 2025), the intermediate task of manipulating scene graphs has received far less attention. A handful of methods embed graph edits into their pipelines (Zhai et al., 2023; Dhamo et al., 2021; Chen et al., 2020a; Hu et al., 2022), but none explicitly handle the semantic conflicts that can arise during editing. However, even a simple change, such as a new object addition, requires more than naively inserting a node: one must also infer its plausible relationships with every existing node to avoid breaking commonsense or spatial consistency ( Figure 1(a)). Likewise, modifying a single edge can ripple through the graph, invalidating inter-object dependencies and yielding illogical configurations ( Figure 1(b)). However, methods for detecting and resolving these inconsistencies remain under-explored.

In this work, we propose **SG-Tailor**, an autoregressive model designed to tackle the intricacies of scene graph manipulation. SG-Tailor operates by predicting the relationship between any two nodes in the context of the existing (partial) scene graph. This capability allows the model to infer reasonable edges for newly added nodes while ensuring that edge modifications do not introduce inconsistencies.

When introducing a new node, SG-Tailor queries the node alongside existing nodes of the partial scene graph to infer their relationships, as depicted in Figure 1 (a). This ensures seamless integration of new nodes into the current scene graph structure. For edge modification, the model employs a novel *Cut-And-Stitch* strategy. First, SG-Tailor isolates the subject node from the graph by cutting off all linked edges. Then it infers and "stitches" all relationships conditioned on the rest of the graph, thereby removing all possible conflicts in the graph. In such a way, we bypass the computational complexity of detecting and resolving relationship conflicts, particularly in densely connected graphs.

We validate SG-Tailor through extensive experiments across diverse benchmarks, where we outperform a traditional message-passing baseline and the state-of-the-art open- and closed-source large language models, with and without chain-of-thought prompting, as well as a finetuned LLM baseline. Results demonstrate the robustness of SG-Tailor and its flexibility as a plug-in module for downstream tasks, such as scene generation and robotic manipulation.

Our contributions are summarized as follows:

**1**. We reveal the overlooked problems of scene graph manipulation, highlighting the importance of maintaining semantic coherence during node and edge modifications.

**2**. We propose *SG-Tailor*, an autoregressive model for robust scene graph manipulation capable of commonsense-aware relationship reasoning and conflict solving.

**3**. We demonstrate that SG-Tailor significantly outperforms existing competitors on diverse benchmarks and proves its practical effectiveness as a plug-in module for downstream application tasks.

## 2 RELATED WORK

**Scene Graphs.** Scene graphs, as symbolic and semantic representations (Johnson et al., 2015; Krishna et al., 2017; Armeni et al., 2019), can be obtained from texts (Zhao et al., 2023), 2D images (Xu et al., 2017; Zellers et al., 2018; Qi et al., 2019; Herzig et al., 2018), 3D geometry (Koch et al., 2024; Rosinol et al., 2021) and 4D data (Yang et al., 2023) for spatial and temporal understanding. Scene graphs can facilitate various tasks, including retrieval (Johnson et al., 2015), generation (Johnson et al., 2018), and VQA (Teney et al., 2017). More embodied applications include robotic manipulation (Zhai et al., 2024a; Jiang et al., 2024), and mobile navigation (Rana et al., 2023). While most works focus on how to embed information into the graphs and how to generate concrete content from scene graphs, to the best of our knowledge, two works (Chen et al., 2020a; Hu et al., 2022) explicitly focused on scene graph manipulation. (Chen et al., 2020a) formulates graph editing as an RL problem, where actions add/remove triplets to minimize graph edit distance to a target. Demonstrated on 2D

image graphs, but suffers from high action-space complexity. (Hu et al., 2022) grows a scene graph by iteratively adding nodes and edges via learned expansion rules. This approach is effective for graph completion, but does not explicitly handle semantic conflicts arising from edits. The closest work to ours is SGNet (Zhou et al., 2019), which predicts objects taking the contextual scene graph into account through message passing, but it also uses object location information. In contrast, our framework focuses on reasoning inter-object relationships based on textual information without explicit geometric cues.

**Autoregressive Models and Their Use on Graphs.** Autoregressive models sequentially predict the next component based on previous inputs. In earlier years, they have shown the possibility of generating images in a row-by-row, raster-scan manner (Van den Oord et al., 2016; Chen et al., 2020b). Recently, autoregressive models have dominated natural language processing (NLP), serving as a crucial component of Large Language Models (LLMs) (Touvron et al., 2023; Brown et al., 2020; Alayrac et al., 2022; Guo et al., 2025; Liu et al., 2023).

Beyond text generation, autoregressive models are used in graph structure data to capture the graph context (Dai et al., 2020; Liao et al., 2020). Since scene graphs can be easily represented by natural textual information, we use autoregressive models to treat scene graph manipulation as a next-token prediction task inspired by LLMs.

## 3 PROBLEM FORMULATION

The **scene graph manipulation problem** exemplifies the physical rearrangements of scenes. However, naively inserting or changing nodes often breaks commonsense or spatial consistency. In this section, we (1) define a class of "reasonable" scene graphs constrained by learned commonsense and spatial constraints, and (2) show how all typical graph edits — adding, removing, or changing relationships — can be reduced to *Cut* and *Stitch* procedures.

### 3.1 SCENE GRAPH AND ITS TRIPLET DESCRIPTION

The scene graph used in this work is officially defined as the triplet description of a visual scene similar to (Li et al., 2024). Given a visual scene $S \in \mathcal{S}$, such as an image or a 3D mesh, its scene graph is a set of triplets $G_S \subseteq \mathcal{O}_S \times \mathcal{P}_S \times \mathcal{O}_S$, where $\mathcal{O}_S$ is the object set, and $\mathcal{P}_S$ is the relation set. Each object $o_{S,k} \in \mathcal{O}_S$ has a semantic label $l_{S,k} \in \mathcal{L}_O$ ($\mathcal{L}_O$ is the semantic label set), where $k \in \{1, \ldots, |\mathcal{O}_S|\}$. Each relation

$$p_{S,i \to j} \in \mathcal{P}_S \subseteq \mathcal{P}$$

is the core form of a visual relationship triplet

$$t_{S,i \to j} = \langle o_{S,i}, \, p_{S,i \to j}, \, o_{S,j} \rangle \; \in \; G_S, \tag{1}$$

with $i \neq j$.

In the terms used in graph theory, a scene graph is a directed graph with two types of nodes: object and relation. However, for the convenience of semantic expression, we conventionally refer to a node of a scene graph as an object with its semantic label, while the relation is called an edge.

**Definition**. A scene graph is a directed, labeled multigraph

$$G = (V, E), \tag{2}$$

where each node $v \in V$ is an object in the scene with its semantic label, and each edge $e \in E$ is the relation with the start node, end node, and its semantic label. We use $\hat{\mathcal{G}}$ to denote the set of all scene graphs.

### 3.2 REASONABLE SCENE GRAPH

Based on the definition of the set of scene graph $\mathcal{G}$, we denote the set of Reasonable Scene Graphs, or the set of scene graphs that do not violate human intuitions, as $\hat{\mathcal{G}}$. We summarize some empirically observed rules for $\hat{\mathcal{G}}$ here: (1) Reasonable scene graphs respect commonsense constraints. This group of constraints captures human intuition in object placement (e.g., a nightstand typically appears beside a bed, not beside a kitchen table). (2) Reasonable scene graphs respect spatial constraints. This

group of constraints forbids logically impossible arrangements (e.g., a chair cannot be simultaneously at the left and right of the same table).

In this work, we consider the scene graphs that do not satisfy these two rules as not reasonable and containing conflicting information.

**Optimal Substructure.** In real-life observation, moving an object of interest would intuitively not affect the rest of the scene composition. We summarize this intuition as the following property that supports our Cut–and–Stitch method. For any reasonable graph $G = (V, E) \in \hat{\mathcal{G}}$ and any subset $S \subseteq V$, the induced subgraph

$$G[S] = (S, E_S), \text{ where } E_S = \{ e_{i \to j} \in E \mid i, j \in S \} \tag{3}$$

is also reasonable and lies in $\hat{\mathcal{G}}$.

This property holds for all relationship types of all datasets used in the experiments. Please refer to the appendix for the complete list of those relations.

### 3.3 SCENE GRAPH MANIPULATION

We formally define *scene graph manipulation*, the process of modifying $G \in \mathcal{G}$ to $G' \in \hat{\mathcal{G}}$, as a series of graph-level operations that mirror a user's adjustments on a scene. These graph-level operations include: *Node Addition*, *Node Removal*, and *Edge Change*. Although these fundamental operations—adding objects, removing objects, and modifying relationships—are conceptually simple, each may result in conflicts and requires precise management. We define the three graph-level operations as follows.

**Node Addition.** We denote a new node as $v$. Define $\bar{e} \subseteq \{ e_{v \to v_i} : v_i \in V \}$ as the set of edges between $v$ and each $v_i$. Node addition is then formulated as constructing a new graph

$$G' = \left( V \cup \{v\}, E \cup \bar{e} \right) \in \hat{\mathcal{G}}. \tag{4}$$

**Node Removal.** Let the node to be removed be $v \in V$, and the associated edges are $\bar{e} \subseteq \{ e_{v \to v_i} : v_i \in V \setminus \{v\} \}$, where $v_i$ are the connected nodes. Thus, it is formulated as constructing a new graph

$$G' = \left( V \setminus \{v\}, E \setminus \bar{e} \right) \in \hat{\mathcal{G}}. \tag{5}$$

**Edge Change.** Let the node of interest be $v \in V$. Denote by $\bar{e}' \subseteq E$ the set of edges originally incident to $v$ and by $\bar{e} \subseteq \{ e_{v \to v_i} : v_i \in V \setminus \{v\} \}$ the set of predicted edges connecting $v$ to the other nodes. The edge change operation is formulated as constructing a new graph

$$G' = \left( V, (E \setminus \bar{e}') \cup \bar{e} \right) \in \hat{\mathcal{G}}. \tag{6}$$

## 4 SG-TAILOR METHODOLOGY

### 4.1 THE CUT-AND–STITCH STRATEGY

To address the above-mentioned three graph-level operations of scene graph manipulation, we introduce the novel *Cut-And-Stitch* strategy, which consists of the *Cut-Step* and the *Stitch-Step*: the **Cut-Step** involves the search and deletion of all edges related to the node of interest, effectively isolating the node. The **Stitch-Step** involves accurate inter-object reasoning and the establishment of relationships between the new node and all existing nodes, which is crucial for the resulting graph to follow the constraints in $\hat{\mathcal{G}}$.

With these two steps, the graph-level operations can be modelled as:

**1. Node Addition**: a *Stitch-Step* for the target node.

**2. Node Removal**: a *Cut-Step* for the target node.

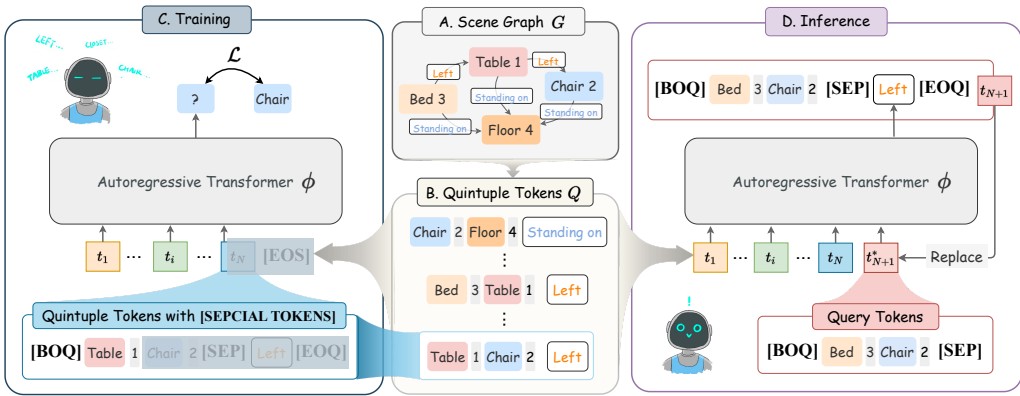

Figure 2: **Training and Inference.** Starting from *A. Scene Graph in its triplet form* $G_S$, we convert $N = |E|$ triplets into a set of *B. Quintuple Tokens* $Q$, resulting in $5N$ tokens. Each token $q_i \in Q$ is then combined with special tokens to form extended quintuple tokens $\widehat{Q}$, with $8N + 1$ tokens in total. During *C. Training*, the model $\phi$ learns to perform next-token prediction on the extended quintuple tokens with the mask attention mechanism. This process runs until it reaches the sequence-end token [**EOS**]. During *D. Inference*, $\phi$ accepts tokens from the existing graph $\left\{ \widehat{Q}_1, ... \widehat{Q}_N \right\}$ from the given graph and query tokens $\widehat{Q}^*_{N+1}$ containing each two of the nodes and special tokens to perform next-relationship prediction. The predicted relationship is integrated into $\widehat{Q}^*_{N+1}$, forming $\widehat{Q}_{N+1}$.

**3. Edge Change**: a *Cut-Step* for the target node followed by the introduction of the target relation, as well as a *Stitch-Step*. This is also referred to as *Cut-And-Stitch*.

## 4.2 FROM TRIPLETS TO TOKENS

We formulate inter-object relationship reasoning as an **autoregressive sequence generation** task. To enable autoregressive modeling based on a transformer architecture, we first convert every subject, object, and predicate, as defined in Equation 1, into tokens. Specifically, to distinguish nodes of the same category, we decompose each of the subjects and objects into two tokens, each representing the class label and instance ID. Together with the token that represents the predicate, there are 5 tokens per triplet, forming the quintuple $Q = t_s^{\text{cls}}, t_s^{\text{ind}}, t_o^{\text{cls}}, t_o^{\text{ind}}, t_{\text{p}}$, where s, o, p denote subject, object, and predicate, respectively; cls and ind denote class category name and instance index, respectively. We augment these tokens with three special tokens to form an extended representation of each triplet.

$$\widehat{Q} = [\textbf{BOQ}] \; t_s^{\text{cls}} \; t_s^{\text{ind}} \; t_o^{\text{cls}} \; t_o^{\text{ind}} \; [\textbf{SEP}] \; t_{\text{p}} \; [\textbf{EOQ}] \qquad \text{(extended quintuple tokens)}$$

where the special tokens, [**BOQ**] (Begin of Quintuple),[**SEP**] (Separation),[**EOQ**] (End of Quintuple), clearly mark the boundaries between subject, object, and relation segments. We concatenate all quintuples of the scene graph $G$ and form the token sequence $t_G$ with an End-of-Sequence token [**EOS**] at the end (see Figure 2.C). This tokenization strategy preserves categorical information, relation types, and instance information for each relationship.

## 4.3 TRAINING: NEXT-TOKEN LEARNING

We propose **SG-Tailor** to address challenges of scene graph manipulation.

At the core of SG-Tailor lies a decoder-only transformer $\phi$ that is trained to generate the scene graph token sequence $t_G$ one token at a time. During training, $\phi$ is supervised by every token inside of $t_G$ to learn in diverse domains, including subjects, objects, and their predicates. Hence, masking is applied at the individual token level rather than at the level of the entire triplet sequence, as shown in Figure 2.C. In the ablation study subsection 5.5, we demonstrate that it improves performance by providing diverse supervision rather than only focusing on the supervision of relationships.

**Modeling.** Specifically, $\phi$ processes input sequences comprising tokens from previous extended quintuple tokens $\{\widehat{Q}_1, \widehat{Q}_2, \ldots, \widehat{Q}_i\}$ and the incomplete tokens $\widehat{Q}_{i+1}[0 : c], 0 \leq c < |\widehat{Q}_{i+1}|$, where $|\widehat{Q}_{i+1}|$ denotes the length of the token sequence, to predict the next token $\widehat{Q}_{i+1}[c+1]$. Formally, at each prediction step, we have:

$$P\big(\widehat{Q}_{i+1}[c+1] \mid \widehat{Q}_1, \ldots, \widehat{Q}_i, \widehat{Q}_{i+1}[0 : c]\big) = \phi\big(\widehat{Q}_1, \ldots, \widehat{Q}_i, \widehat{Q}_{i+1}[0 : c]\big). \tag{7}$$

This autoregressive modeling enables $\phi$ to capture intricate dependencies at both the entity level and predicate level across the entire sequence, thus enhancing the reasoning capability on inter-object relationships.

**Training Objective.** We adopt a categorical cross-entropy loss across the entire vocabulary to train the model:

$$\mathcal{L} = -\sum_{t=1}^{T} \log \frac{\exp(z_{t,y_t})}{\sum_{w \in \mathcal{V}} \exp(z_{t,w})}, \tag{8}$$

where $T$ is the total number of tokens in the sequence, $\mathcal{V}$ represents the token vocabulary, $y_t$ denotes the ground-truth token at step $t$, and $z_{t,w}$ indicates the logit (unnormalized score) assigned by the model to token $w$ at step $t$. This loss guides the model to accurately predict each token, ensuring the effective autoregressive modeling of scene graph structures.

### 4.4 INFERENCE: NEXT-RELATIONSHIP REASONING

During inference, SG-Tailor performs autoregressive reasoning to predict the inter-object relationship between a query node and another node, conditioning each prediction on the previously established graph connections. Given token sequences $\widehat{Q}_{1:i}$ representing all the complete quintuples in the graph, and the tokens of the subject and object of the quintuple of interest given by the incomplete quintuple:

$$\widehat{Q}^*_{i+1} = \Big([\mathbf{BOQ}], t^{\mathrm{cls}}_{s_{i+1}}, t^{\mathrm{ind}}_{s_{i+1}}, t^{\mathrm{cls}}_{o_{i+1}}, t^{\mathrm{ind}}_{o_{i+1}}, [\mathbf{SEP}]\Big). \tag{9}$$

The model computes the conditional probability of the $i + 1$th predicate token

$$P(t_{p_{i+1}} \mid \mathrm{concat}(\widehat{Q}_{1:i}, \widehat{Q}^*_{i+1})) = \phi(\widehat{Q}_{1:i}, \widehat{Q}^*_{i+1}). \tag{10}$$

This formulation enables the model to capture the intricate interdependencies among objects, ensuring that each new prediction respects the existing graph structure.

During every inference step, the subject and object tokens are queried as described in Figure 2.D. By the end of the inference, the scene graph $G' \in \widehat{\mathcal{G}}$ is built from the tokens.

## 5 EXPERIMENTS

### 5.1 DATASETS

We evaluate our method quantitatively and qualitatively on three datasets, 3RScan (Wald et al., 2019), 3D-FRONT (Fu et al., 2021), and SceneVerse (Jia et al., 2024) with different motivations. Both 3RScan and 3D-FRONT are widely used as benchmarks for scene-graph-based 3D scene generation (Dhamo et al., 2021; Zhai et al., 2023), which allows evaluation of our method on the 3D scene manipulation downstream task in section 6. Experiments on SceneVerse37K show the scalability of our method in large-scale scenarios. We also report the user preference ranking of our method as a user perceptual study in the appendix.

### 5.2 IMPLEMENTATION DETAILS

We use Llama (Touvron et al., 2023) layers as the autoregressive transformer in our model, with a hidden size of 768 and 12 attention heads. We pad the scene graph sequence to the context length 1024 and use a cosine learning rate scheduler with an initial learning rate of $5 \times 10^{-4}$ and a weight decay of $1 \times 10^{-2}$. The batch size is set to 16 in all our experiments, and we train our models for 50 epochs, employing early stopping. We predict all relationships using nucleus sampling (Holtzman et al., 2020) with a p-value of 0.7. We augment our data by randomly shuffling each scene graph three times.

## 5.3 Evaluation Metrics

**Ranking-based Evaluation Metrics.** In the evaluation of scene graphs, there may be multiple valid ground truth labels that differ from the predicted label. However, since it is logical to assume the label present in the dataset should appear high in the probability distribution predicted by a well-trained model, we adopt ranking-based metrics (Bordes et al., 2013). This enables evaluation of our method without additional labeling.

*Mean Rank (MR)* calculates the average ranking position of correct predictions, where lower ranks indicate better performance. *Mean Reciprocal Rank (MRR)* averages the reciprocal rank of the first correct prediction, emphasizing early correct answers and penalizing lower-ranked, delayed correct predictions. *Hit@K* computes the proportion of queries where the correct answer appears within the top K predictions. Please refer to the appendix for the equations for these metrics.

**Cycle Rates.** Taking the intuition that the presence of a cycle signifies a spatial contradiction within the scene graph, we identify spatial conflicts through Algorithm 1 in the appendix, a simple graph loop detection algorithm based on depth-first search. This experiment is conducted among the naive approach, Llama-3.3-70B-Instruct, MPNN baseline, and our approach by modifying spatial relations (left, right, front, behind) to generate new scene graphs.

## 5.4 Baselines

Since we introduce scene graph manipulation as a novel task, we construct three types of baselines, each representing the naive evaluation procedure, traditional approaches for learning graph data, and content generation with modern LLMs.

**Naive Manipulation Baseline (Naive).** Following (Zhai et al., 2023; 2024b), we compare our method to this naive baseline where no reasoning is performed. **Message-Passing Neural Network (MPNN) Baseline.** Inspired by (Zhou et al., 2019), we built an MPNN baseline for the scene graph manipulation task. SGNet is a method that employs the message-passing mechanism for the prediction of the likelihood over classes at a query location. We modified the architecture of SGNet to encode class labels and use them for message-passing, and built two MLP decoders for the node addition and edge change tasks, respectively. **LLM baselines.** We compare our approach with state-of-the-art large-language models, including open-source and proprietary models. The model specifications and results are summarized in Table 1, 2, and 5. As a practical compromise under our limited budget, we use the open-source Llama-3.3-70B-Instruct (Grattafiori et al., 2024) for experiments on the scene-graph consistency metrics. In addition, we finetuned Gemini-2.5 (Google DeepMind, 2025) on the 3D-FRONT dataset. All prompt templates and details on the exact finetuning procedure are provided in the appendix.

## 5.5 Ablation Study

We also evaluate our method against two other variants based on the same dataset and evaluation metrics in Table 1. Specifically, we test the following variants: (1) **SG-Tailor (GPT-2).** We train the GPT-2 layers (Radford et al., 2019) of the same depth and width on the same next token prediction task. (2) **SG-Tailor (Next-Rel)** We train SG-tailor with loss label masks on all subjects and objects, effectively guiding the model to only learn to predict the predicates. Our method outperforms all these variants.

## 5.6 Quantitative Results

We summarize the performance of our method and the baselines in the following table. Please refer to the appendix for more results:

On **3RScan**, our method achieves notable improvements over the MPNN baseline. These results show that our method is capable of capturing the details in relatively easier scenes.

On **SceneVerse37K**, our full SG-Tailor model performs better than the baseline on this large-scale and challenging dataset on all metrics with a **5.52%** relative increase on Hit@1, demonstrating better generalizability. We argue that MPNNs have less generalization ability.

| Method | Dataset | MR ↓ | MRR ↑ | Hit@1 ↑ | Hit@3 ↑ | Hit@10 ↑ |
|---|---|---|---|---|---|---|
| MPNN | 3RScan (Wald et al., 2019) | 3.987 | 0.572 | 0.398 | **0.683** | 0.941 |
| Ours | | **3.764** | **0.624** | **0.451** | 0.681 | **0.953** |
| MPNN | SceneVerse37K (Jia et al., 2024) | 5.922 | 0.367 | 0.199 | 0.426 | 0.882 |
| Ours | | **5.623** | **0.374** | **0.210** | **0.507** | **0.92** |
| GPT-4.1-nano (OpenAI, 2025a) | 3D-FRONT (Fu et al., 2021) | 8.010 | 0.226 | 0.076 | 0.197 | 0.664 |
| GPT-4.1 (OpenAI, 2025a) | | 5.697 | 0.389 | 0.235 | 0.402 | 0.834 |
| o4-mini (w/ CoT) (OpenAI, 2025b) | | 4.948 | 0.384 | 0.203 | 0.420 | 0.949 |
| Gemini-2.5 (Google DeepMind, 2025) | | 4.646 | 0.360 | 0.152 | 0.392 | 0.975 |
| Gemini-2.5-finetuned (Google DeepMind, 2025) | | 4.533 | 0.382 | 0.210 | 0.435 | 0.981 |
| MPNN | | 4.103 | 0.486 | 0.271 | 0.593 | 0.943 |
| Ours (GPT-2) | | 4.113 | 0.472 | 0.273 | **0.604** | 0.947 |
| Ours (Next-Rel) | | 4.113 | 0.476 | 0.282 | 0.603 | 0.951 |
| Ours | | **3.613** | **0.498** | **0.305** | 0.591 | **0.983** |

Table 1: Performance metrics across models and datasets. Best results are in bold.

On **3D-FRONT**, our final configuration delivers an **11.9%** reduction in MR (from 4.103 to 3.613) and achieves superior Hit@1 (0.305 vs. 0.271) and Hit@10 (0.983 vs. 0.943) scores compared to MPNN. Our method outperforms the MPNN baseline as well as the latest LLMs, demonstrating the ability to deal with complex scene relationships. Notably, our method also achieves better overall results than the finetuned version of Gemini-2.5.

**Cycle Rates.** As shown in Table 2, more than 30% of the scene graphs generated by the naive approach contain significant contradictions, while our method effectively mitigates these inconsistencies, achieving a remarkably low total cycle rate of just 1%. We infer that the exceptionally high cycle rate observed for the Llama-3.3 baseline is likely due to LLMs being prone to noise (Kim et al., 2024).

| Method | right cycle ↓ | front cycle ↓ | total ↓ |
|---|---|---|---|
| Naive | 19.19% | 19.73% | 38.92% |
| Llama-3.3 (Grattafiori et al., 2024) | 34.48% | 31.03% | 62.07% |
| MPNN | 7.37% | **0.0** | 7.37% |
| SG-Tailor (Ours) | **1.05%** | **0.0** | **1.05%** |

Table 2: **Cycle rate**. The percentage of scene graphs that have either a right or front cycle. Best in bold.

Overall, these quantitative results demonstrate that conditioning on the entire scene graph context improves the ranking accuracy of relationship predictions, and yields more coherent scene graphs. The consistent performance gains across different datasets and metrics underscore the robustness and effectiveness of our approach.

# 6    DOWNSTREAM APPLICATIONS AND FURTHER QUANTITATIVE RESULTS

We show the ease of use of SG-Tailor in the scene manipulation settings. A further application case in robotics can be found in the appendix.

Operating entirely at the scene graph level, our model integrates seamlessly with existing scene-graph-based 3D scene generation frameworks (Dhamo et al., 2021; Zhai et al., 2023; 2024b). We train SG-Tailor on the 3D-FRONT dataset, the same dataset used for the downstream module Graph-to-3D, and perform scene graph manipulation before feeding the graph into Graph-to-3D for 3D rendering. We compare the rendered manipulation results of our method with those of the MPNN baseline, GPT-4.1, and Llama-3. See Figure 3.

Our method effectively captures the global context, resulting in scene graphs that are both accurate and coherent. In contrast, alternative methods struggle to enforce object constraints and predict accurate relationships. In addition, we conducted a user study to qualitatively evaluate the scene graph after manipulation and addition. More results are available in the appendix.

## 6.1    SCENE GRAPH CONSTRAINTS

Following (Zhai et al., 2023; 2024b), we measure the spatial accuracy of the manipulated scenes, with the results summarized in Table 3.

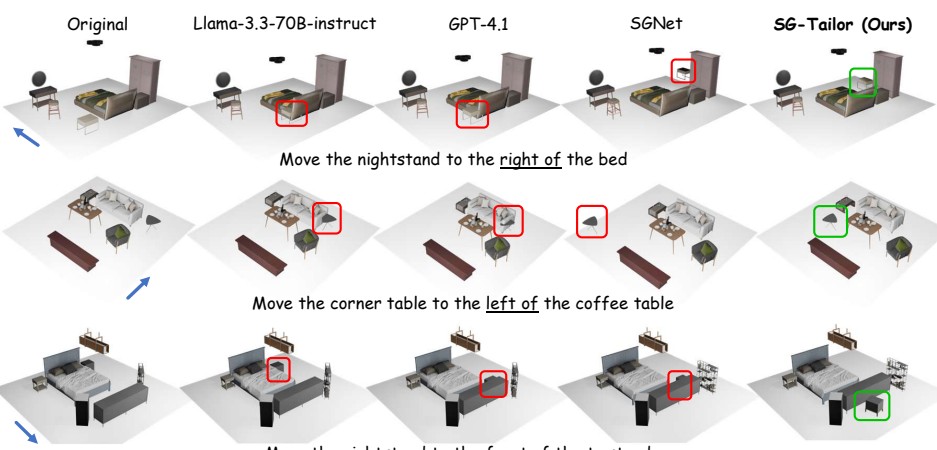

Figure 3: **Qualitative comparison.** We assess the quality of scene graph manipulation by generating the corresponding scenes using the Graph-to-3D model (Dhamo et al., 2021). The blue arrow indicates the front direction of the scene.

| Method | left/right | front/behind | bigger/smaller | taller/shorter | close by |
|---|---|---|---|---|---|
| Naive | 0.93 | 0.93 | 0.97 | 0.95 | 0.67 |
| Llama-3.3 | 0.92 | 0.93 | 0.98 | **0.97** | **0.68** |
| MPNN | 0.97 | **0.97** | **0.98** | **0.97** | 0.55 |
| SG-Tailor (Ours) | **0.98** | 0.97 | 0.96 | **0.97** | **0.68** |

Table 3: Comparison of Methods on Different Relation Types. The total accuracy is computed as the mean over the individual edge class.

## 7    LIMITATIONS

Our current pipeline leverages Graph-to-3D (Dhamo et al., 2021) for downstream scene synthesis. This introduces two main limitations: **Limitation on relationship types.** For compatibility with Graph-to-3D, SG-Tailor's relationship vocabulary must match the fixed set of predicates supported by Graph-to-3D. Other relation types are not to be visualized. But the effectiveness of SG-tailor is demonstrated in other experiments. **Imperfect graph–render consistency.** Due to modeling and rendering constraints in Graph-to-3D, generated 3D outputs can occasionally violate the input scene graph (e.g., slight misplacements or missing objects). As future work, one could integrate reinforcement-learning–based fine-tuning of SG-Tailor (e.g., optimizing a renderer-in-the-loop reward) to better bridge the gap between predicted graphs and final 3D fidelity.

## 8    CONCLUSION

We have introduced **SG-Tailor**, a decoder-only autoregressive model for scene graph editing. By framing every manipulation (addition, removal, or modification of edges) as next-token prediction over a vocabulary, SG-Tailor reasons holistically about global context and commonsense constraints. In experiments on the scene graph manipulation task, SG-Tailor consistently outperforms state-of-the-art open- and closed-source large language models, both with and without chain-of-thought prompting, as well as a message passing network baseline. This work opens new directions for intelligent and context-aware scene generation, with promising applications in diverse 3D content creation, interactive editing tools, and goal-driven robotic manipulation.

## LLM USAGE

ChatGPT was used exclusively as a tool for grammar checking and language polishing. The research ideas, experimental design, analyses, and conclusions were fully developed and carried out by the authors. At no stage were ChatGPT or other LLMs employed for ideation, technical content generation, or methodological purposes.

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

# APPENDIX

## A ROBOTIC MANIPULATION

To further show the flexibility and plug-and-play of SG-Tailor, we bring SG-Tailor to tabletop environments for robotic manipulation tasks. We train our method on the SG-Bot dataset (Zhai et al., 2024a) to evaluate how our method facilitates scene-graph-based robotic manipulation. While SG-Bot excels at generating precise target configurations, it often encounters difficulties rearranging objects when conflicts emerge in the scene graph following relationship updates. As illustrated in Figure 4. This integration produces more coherent and context-aware scene representations, significantly enhancing SG-Bot's planning accuracy and execution efficiency. Additional qualitative results are presented in the appendix.

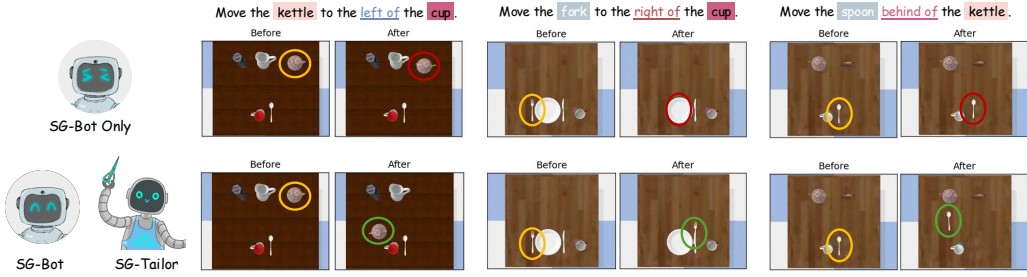

Figure 4: **Qualitative comparison of SG-Bot w/ and w/o SG-Tailor.** We show three examples of SG-Tailor facilitating the robotic manipulation tasks. More examples can be found in the appendix.

## B  RANKING METRICS

As discussed in the main paper, we evaluate our method with ranking-based metrics: mean rank, mean reciprocal rank, and Hits@k. The detailed definition is as follows:

**Mean Rank (MR)**

$$\mathbf{MR} = \frac{1}{N} \sum_{i=1}^{N} \text{rank}_i. \tag{11}$$

**Mean Reciprocal Rank (MRR)**

$$\mathbf{MRR} = \frac{1}{N} \sum_{i=1}^{N} \frac{1}{\text{rank}_i}. \tag{12}$$

**Hit@k**

$$\mathbf{Hit@k} = \frac{1}{N} \sum_{i=1}^{N} \mathbb{1}\{\text{rank}_i \leq k\}. \tag{13}$$

where $\mathbb{1}$ is the indicator function that returns 1 if the ground truth label for the i-th query appears within the top k predictions, and 0 otherwise.

## C  CYCLE DETECT ALGORITHM

We identify spatial conflicts through Algorithm 1, which is a graph loop detection algorithm based on depth-first search (DFS). Specifically, we convert left and behind triplets into their right and front counterparts and then detect cycles in the right and front relationships.

---

**Algorithm 1** DFS-based Cycle Detection in a Directed Graph

---

1: **procedure** DETECTCYCLE($G$)
2:     $visited \leftarrow \emptyset$
3:     $recStack \leftarrow \emptyset$
4:     **for** each vertex $v$ in $G$ **do**
5:         **if** $v \notin visited$ **then**
6:             **if** DFS($v, visited, recStack, G$) **then**
7:                 **return true**                                                     ▷ Cycle detected
8:             **end if**
9:         **end if**
10:     **end for**
11:     **return false**                                                            ▷ No cycle found
12: **end procedure**

---

## D TABLE 1 IN FULL: ADDITIONAL LLM EVALUATION RESULTS ON 3RSCAN AND SCENEVERSE

This section presents an extended version of Table 1, where large language model baselines are also included for the datasets 3RScan and SceneVerse100. Due to the budget limit, the comparison on SceneVerse is conducted on this small subset of size 100 to demonstrate our method's generalization ability to this dataset.

| Method | Dataset | MR ↓ | MRR ↑ | Hit@1 ↑ | Hit@3 ↑ | Hit@10 ↑ |
|---|---|---|---|---|---|---|
| GPT-4.1-nano (OpenAI, 2025a) | | 7.240 | 0.305 | 0.122 | 0.257 | 0.718 |
| GPT-4.1 (OpenAI, 2025a) | | 5.672 | 0.336 | 0.245 | 0.399 | 0.834 |
| o4-mini (OpenAI, 2025b) (w/ CoT) | | 4.525 | 0.354 | 0.201 | 0.430 | 0.944 |
| Gemini-2.5-flash (Google DeepMind, 2025) | 3RScan (Wald et al., 2019) | 4.610 | 0.350 | 0.171 | 0.502 | 0.961 |
| MPNN | | 3.987 | 0.572 | 0.398 | **0.683** | 0.941 |
| Ours | | **3.764** | **0.624** | **0.451** | 0.681 | **0.953** |
| MPNN | | 5.922 | 0.367 | 0.199 | 0.426 | 0.882 |
| Ours | SceneVerse37K (Jia et al., 2024) | **5.623** | **0.374** | **0.210** | **0.507** | **0.921** |
| GPT-4.1-nano (OpenAI, 2025a) | | 8.711 | 0.196 | 0.066 | 0.182 | 0.651 |
| GPT-4.1 (OpenAI, 2025a) | | 6.373 | 0.372 | 0.237 | 0.511 | 0.824 |
| o4-mini (OpenAI, 2025b) (w/ CoT) | | 5.338 | 0.370 | 0.201 | 0.389 | **0.916** |
| Gemini-2.5-flash (Google DeepMind, 2025) | SceneVerse100 (Jia et al., 2024) | **5.141** | 0.298 | 0.126 | 0.319 | 0.873 |
| Ours | | 5.819 | **0.371** | **0.207** | **0.521** | 0.899 |
| GPT-4.1-nano (OpenAI, 2025a) | | 8.010 | 0.226 | 0.076 | 0.197 | 0.664 |
| GPT-4.1 (OpenAI, 2025a) | | 5.697 | 0.389 | 0.235 | 0.402 | 0.834 |
| o4-mini (OpenAI, 2025b) (w/ CoT) | | 4.948 | 0.384 | 0.203 | 0.420 | 0.949 |
| Gemini-2.5-flash (Google DeepMind, 2025) | | 4.646 | 0.360 | 0.152 | 0.392 | 0.975 |
| Gemini-2.5-finetuned (Google DeepMind, 2025) | 3D-FRONT (Fu et al., 2021) | 4.533 | 0.382 | 0.210 | 0.435 | 0.981 |
| MPNN | | 6.253 | 0.332 | 0.201 | 0.417 | 0.859 |
| MPNN | | 4.103 | 0.486 | 0.271 | 0.593 | 0.943 |
| Ours (GPT-2) | | 4.113 | 0.472 | 0.273 | **0.604** | 0.947 |
| Ours (Next-Rel) | | 4.113 | 0.476 | 0.282 | 0.603 | 0.951 |
| Ours | | **3.613** | **0.498** | **0.305** | 0.591 | **0.983** |

Table 4: Performance metrics across models and datasets. Best results are in bold.

## E SCENE GRAPH EXAMPLES USED IN SEC. 6.1

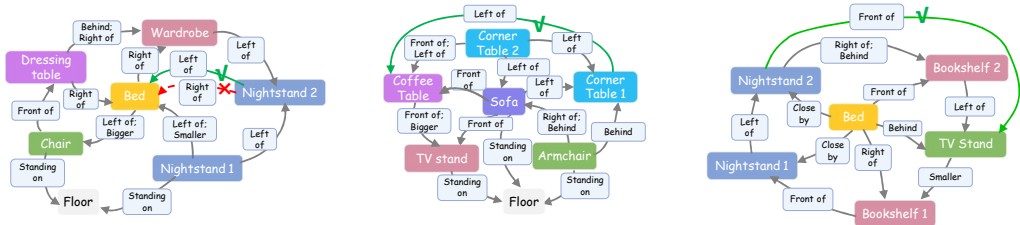

Figure 5: The corresponding scene graphs of the scenes visualized in Fig. 3. Green lines represent the desired new relationships. SG-Tailor first removes all existing edges connected to the target object (red line is an example of edges to be cut) and then infers its new relationships with other objects.

# F   LLM RANKING METRICS PROMPT

We provide the prompts used to calculate the performance metrics on the 3DFront (Fu et al., 2021) dataset reported in Table 1. Note that the relationship types are limited to the possible relationships in the 3DFront (Fu et al., 2021) dataset. Similar prompts are used in the evaluation of the other datasets (Wald et al., 2019; Jia et al., 2024).

```
You are given a partial scene-graph triplet and must predict the missing
    spatial relationship code.

Input:
One triplet per line in the form:
<subject> <object> <relationship>

On all but the last line, <relationship> is one of the codes from 1 to 15
    (see mapping below).

On the final line, only <subject> and <object> appear; your job is to
    rank all 15 codes for that pair.

Relationship mapping:
1 left
2 right
3 front
4 behind
5 close_by
6 above
7 standing_on
8 bigger_than
9 smaller_than
10 taller_than
11 shorter_than
12 symmetrical_to
13 same_style_as
14 same_super_category_as
15 same_material_as

Output:
A single line containing 15 index numbers, from 1 to 15, ordered from
    most to least likely for the missing relationship, separated by
    spaces. Conform to spatial common sense and constraints.
Do not include any additional text.

Example:
Input:
chair_1 floor_1 standing_on
desk_1 floor_1 standing_on
chair_1 desk_1 left
chair_2 desk_1 right
chair_2 floor_1

Output:
7 6 5 1 2 3 4 9 10 13 15 14 8 11 12
```

# G LLM MANIPULATION PROMPT

We provide the prompts that we use for the edge change task with the model Llama-3.3-70B-Instruct (Touvron et al., 2023). Note that the types of relationships used in this experiment are limited to those available in Graph-to-3D (Dhamo et al., 2021).

```
[Context]
You are a helpful assistant whose task is
to manipulate a node in a scene graph.
The scene graph is represented as
triplets in the following format:
subject object relationship

Possible relationships are:
left
right
front
behind
standing_on
bigger_than
smaller_than

Input Format:
First line: The incomplete triplet of the node to be set.

Subsequent lines: Existing scene
graph triplets (one per line).

Instructions:
Output Requirements:
Respond only with the complete scene
graph triplets after adding new triplets.

Do not include any explanation or commentary in your output.

Removing Triplets:
Skip the first triplet, and remove any
triplets that contain the subject
of the first triplet.

Adding Triplets:
Use the subject node of the first triplet
as the subject, and select at most
4 other nodes as objects,
and one of the possible relationships,
form at most one triplet for each of the
selected object, and add it to the list.

Only add consistent triplets with the spatial constraints of the existing
    scene graph.
Do not add triplets that are
already present in the scene graph.
Do not add triplets that are contradictory
to the existing scene graph.
Do not add triplets that are redundant.

Spatial Relationships:
Ensure that the updated scene graph
has no contradictions in
spatial relationships.
(A contradiction is defined as two or more
triplets that imply mutually exclusive
spatial configurations.)

Output Format:
```

```
Your response should include the
entire updated scene graph,
in the exact order specified
by the input plus any new valid triplets.
Do not include any extra text, formatting,
or explanations.
Respond one triplet per line.

Example:
Example Input:
chair_1 desk_1 right
chair_1 floor_1 standing_on
desk_1 floor_1 standing_on
chair_1 desk_1 left
chair_2 desk_1 right
chair_2 floor_1 standing_on

Example Output:
desk_1 floor_1 standing_on
chair_2 floor_1 standing_on
chair_1 desk_1 right
chair_1 floor_1 standing_on
chair_1 chair_2 left
[/Context]
```

## H  STATISTICS OF THE PERCEPTUAL USER STUDY

To compensate for the lack of ground-truth 3D scenes for comparison, we conducted a perceptual study with 30 randomly selected participants. In the edge change task, participants are presented with the original scene and scenes generated by four different methods: naive, MPNN, Llama-3.3, and SG-Tailor. In the node addition part, only the results from Llama-3.3, MPNN, and SG-Tailor are compared. Participants were asked to rank the scenes according to how well the changes in the scenes reflect the perceptual similarity to the provided task description.

The summarized results in Table 5 demonstrate that our method outperforms all others in both node addition and edge change tasks. We show that across two tasks, there is a clear trend favoring our method. Our method, combined with the downstream 3D scene generation module (Dhamo et al., 2021; Zhai et al., 2023; 2024b), provides a solid framework for 3D scene manipulation.

| Method | manipulation↑ | addition↑ |
|---|---|---|
| Naive | 1.72% | – |
| Llama-3.3 (Grattafiori et al., 2024) | 9.48% | 18.97% |
| MPNN | 18.10% | 33.33% |
| SG-Tailor (Ours) | **70.69%** | **47.70%** |

Table 5: **Top-1 rate.** The percentage of participants who consider each method across different tasks. Best in bold.

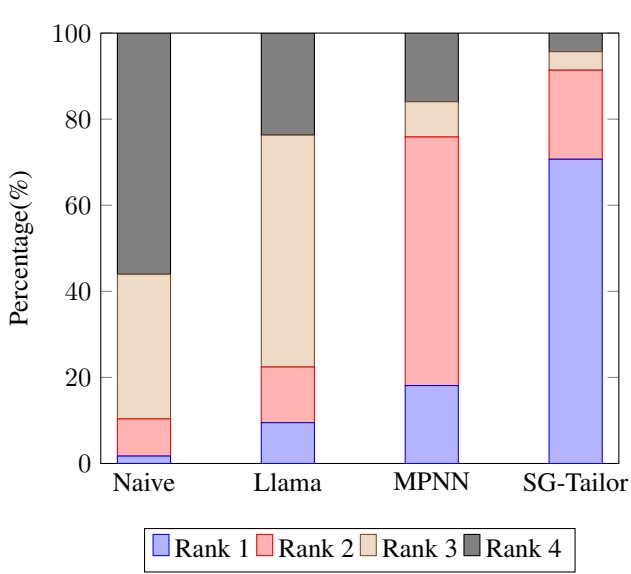

Figure 6: Statistics of the user study in the manipulation task

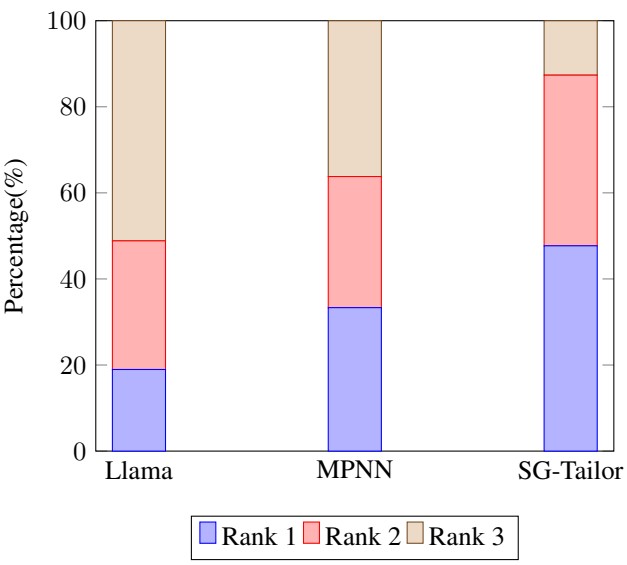

Figure 7: Statistics of the user study in the addition task

# I   ADDITIONAL ROBOTIC MANIPULATION RESULTS

We show additional performance when combining SG-Bot(Zhai et al., 2024a) with SG-Tailor in Figure 8, further showcasing the conflict-resolving ability of SG-Tailor. The stream in the first line marked in red shows that the original SG-Bot can pick up the box, but due to the conflicts in the scene graph, the generation model fails, so the target location is still around the starting pose. In contrast, SG-Tailor can help by resolving the conflicts in the scene graph, so the generative model works again, thereby enabling the rearrangement of objects (see the stream marked in green).

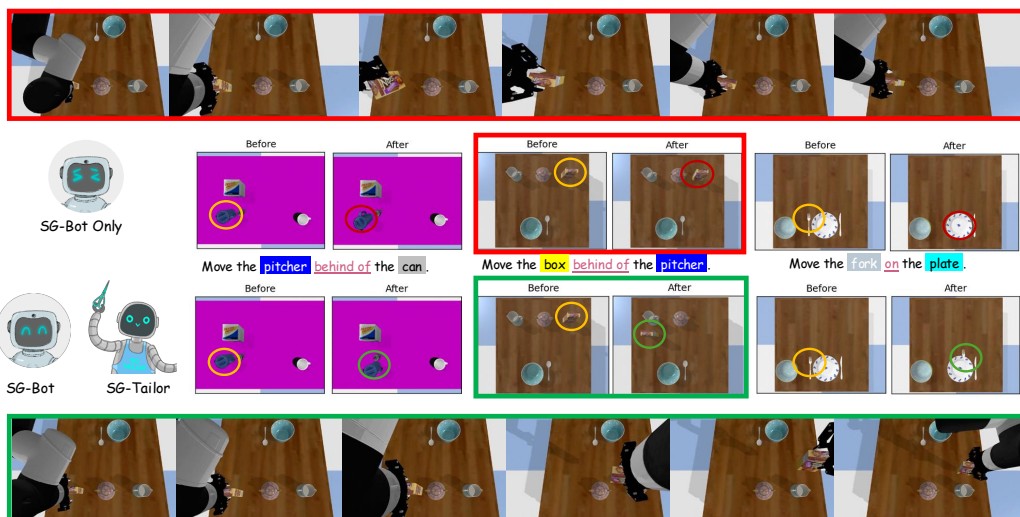

Figure 8: More qualitative comparison of SG-Bot w/ and w/o SG-Tailor.

## J USER STUDY INTERFACE

In this section, we show the interface of the user study in section 6.1, which is conducted to evaluate participants' perceptions of the rendered 3D scene image rankings. The study was structured into two parts, focusing on the node addition and edge change tasks. In each part, participants provided conceptual evaluations of the rankings based on their interpretations. To ensure unbiased responses, the images were presented in a completely randomized order with only minimal instructions provided, as shown in Figure 9.

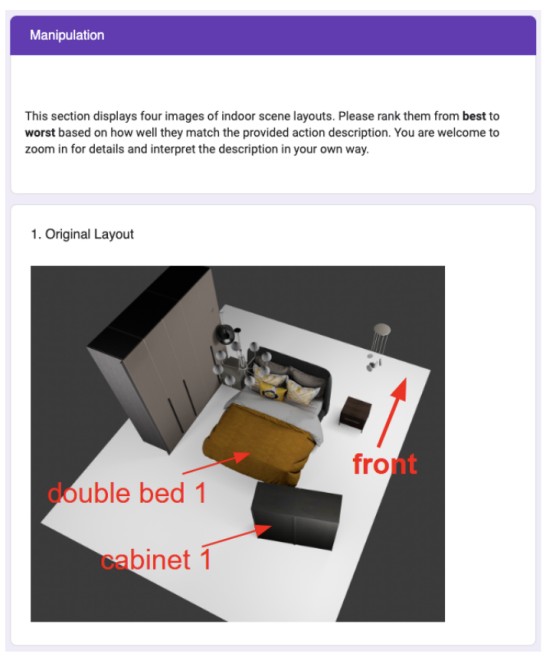

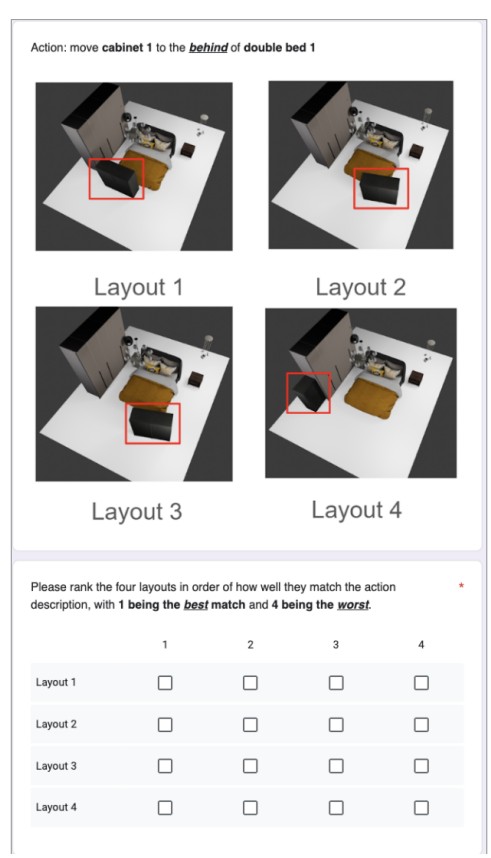

Figure 9: **User interface for the perceptual user study.**

## K FAILURE CASES IN FINAL RENDERING WITH GRAPH-TO-3D, AND ECHOSCENE

In this section, we list rendering examples to show that each of the downstream 3D scene generation modules has its limitations and all suffer from overlap issues. The choice of model is therefore irrelevant for our qualitative evaluation, and we have focused on qualitative evaluation with one of them, Graph-to-3D.

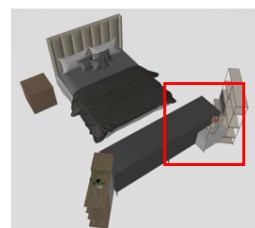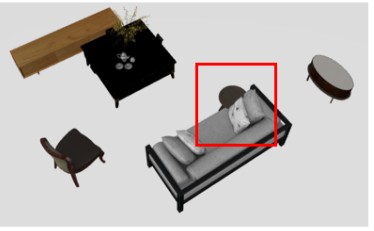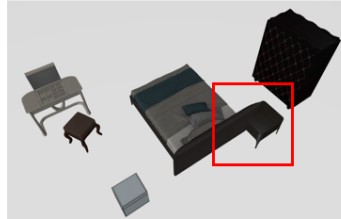

Figure 10: EchoScene (Zhai et al., 2024b) suffers from overlap issues even if the input scene graph is correct.

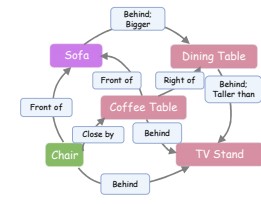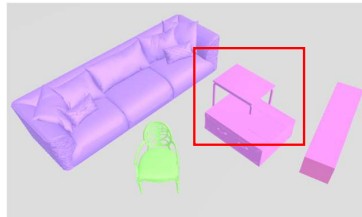

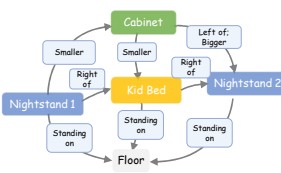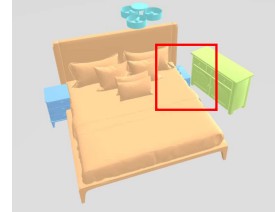

Figure 11: Graph-to-3D (Dhamo et al., 2021) suffers from overlap issues even if the input scene graph is correct.

## L EXAMPLE OF RELATIONSHIP CONFLICT GENERATED BY GPT-4.1

We present an example result of GPT-4.1 in the scene graph manipulation task in Sec. 6.1. Despite being the currently "smartest model" of the GPT family, GPT-4.1 shows an unsatisfactory ability to reason without spatial conflicts.

```
bookshelf_1 pendant_lamp_1 front
... (all other triplets skipped for brevity)
nightstand_1 tv_stand_1 front (desired manipulation relationship)
nightstand_1 floor_1 standing_on
nightstand_1 double_bed_1 right       <<-- contradiction
nightstand_1 double_bed_1 left        <<-- contradiction
nightstand_1 bookshelf_1 behind
nightstand_1 bookshelf_2 behind
nightstand_1 pendant_lamp_1 below
nightstand_1 nightstand_2 same_super_category_as
```

## M RELATIONS PRESENT IN DIFFERENT DATASETS

The types of relations vary across datasets and are summarized in Table 6.

| Relation Type | 3RScan | 3DFront | Sceneverse73k | SG-Bot |
|---|---|---|---|---|
| left of | * | * | * | * |
| right of | * | * | * | * |
| front of | * | * | * | * |
| behind of | * | * | * | * |
| higher than | * | * | | |
| lower than | * | * | | |
| smaller than | * | * | * | |
| bigger than | * | * | * | |
| same as | * | | | |
| close by | | * | | |
| above | | * | * | |
| standing on | * | * | * | * |
| taller than | | * | | |
| shorter than | | * | | |
| symmetrical to | | | | * |
| same style as | | * | | |
| same super category as | | * | | |
| same material as | | * | | |
| support | | * | | |
| embedded in | | * | | |

Table 6: List of All Relation Types (* indicates presence)

## N    FINE-TUNING THE LARGE LANGUAGE MODEL BASELINE

To better evaluate our method against modern large language models, we finetune Gemini-2.5, the best-performing LLM baseline, on a subset of 3DFront through the parameter-efficient finetuning service provided by Google and Vertex AI.

Following the official guidance for fine-tuning, we list the details about dataset construction: for the addition task and the manipulation task, we take 100 randomly selected scene graphs and call them the addition dataset and the manipulation dataset, respectively. Each data sample consists of the system prompt, the user prompt, and the model response. These can be viewed as the instruction, the input, and the expected model output.

The addition dataset is constructed by randomly selecting one node and removing its associated edges in the triplets representation. We construct the expected model response to be this node and its associated edges.

The manipulation dataset is constructed by randomly selecting one node and removing this node, and only keeping one of its associated edges in the triplets representation. The idea is to treat this edge as the desired edge after manipulation. We construct the expected model response to be this node and its associated edges.

System prompts follow the examples provided by Vertex AI, we construct the system prompts:

Addition: You are given a scene object and a scene graph represented as triplets in the order [subject, object, relationship]. You should add the scene object by predicting its relationships with other objects represented by scene graph triplets and respond with the added triplets.

Manipulation: You are given an incomplete scene graph represented as triplets in the order [subject, object, relationship. You should complete the scene graph triplets by predicting all the other triplets associated with the subject.

Training procedure: We randomly took 90 data samples from both datasets and shuffled them together as the training set. We shuffled the other 10 samples from both data sets as the test set.

Inference procedure: To ensure a fair assessment with the other models, it stays the same as the other baseline LLM reported in Sect. 6.1.

