# OpenReview forum: "SG-Tailor: Inter-Object Commonsense Relationship Reasoning for Scene Graph Manipulation"
_ICLR.cc/2026/Conference — Submitted to ICLR 2026_

### Official Review · Reviewer_AvBN · 2025-10-31

**Soundness:** 2
**Presentation:** 2
**Contribution:** 2
**Rating:** 4
**Confidence:** 4

**Summary:**

This work addresses the conflict issues that arise during the graph manipulation steps of generated scene graphs for downstream tasks.

**Strengths:**

Comprehensive description of methods and results.

Demonstrates effectiveness in practical settings and for downstream tasks.

**Weaknesses:**

### 1. Unelaborated Problem Definition
The main problem addressed in the paper is information loss during the manipulation steps. This is acceptable if no method has previously attempted to solve it. However, the third contribution claims to outperform other competitors in these steps, implying solved or solvable issues by competitors. Compared to the competitors, what problems are tackled should be more elaborated.

### 2. Unclear Contribution on Problem Proposal
It seems that the literature may already cover this work, making it important to clarify its novelty. However, Section 3 is vague about whether the formulation itself constitutes a contribution. If it is novel, this should be explicitly stated; if not, the differences from prior work and relevant references should be provided.

### 3. Weak Novelty of Proposed Methods
The method is a straightforward transformer with masked training and does not introduce a particularly novel idea. In particular, the graph neural network community already employs a variety of triplet variants with advanced context-based input variations.

### 3. Issues in Empirical Validation
1) MPNN (2019) is selected as the baseline for comparison without justification. It is one of the earliest works in graph neural networks, and many GNNs have since been developed. Furthermore, it does not support the main argument on the impact of manipulation method proposal, but its generality use in graph building mechanisms, even though the transformer architecture design is not the main contribution of the paper.

2) Downstream tasks for robotics applications present only qualitative results, even though the dataset allows quantitative evaluation in the SG-bot work. Previous results are primarily analytical, and quantitative downstream results are essential to justify the importance of graph manipulation.

**Questions:**

Minor Point

The term “Reasonable Scene Graph” can be confused with “reasoning-enabled scene graph,” and “reasonable” itself is somewhat vague. In my understanding, it simply refers to a scene graph that does not conflict with human knowledge.

---

> ### Author Response · Authors · 2025-11-20
>
> Dear Reviewer AvBN:
> Thank you very much for your effort and time in reviewing our work. Your feedbacks are valuable to us.
>
> > W1: The third contribution claims to outperform other competitors in these steps, implying solved or solvable issues by competitors.
>
> Thank you for pointing this out. The use of the word "competitors" may be a little misleading. As we stated between lines 59-69, there is no other known commonsense-aware method designed to solve scene graph manipulation, and we are the first to reveal the overlooked problems. "Competitors" simply means we compare our method to the naive methods commonly used in other works, as well as some other baselines that are not originally designed for this task but could be adapted to it. We discuss them in section 5.4.
>
> For better clarity, we suggest the following change in wording:
>
> _`Revision`_: *We demonstrate that SG-Tailor significantly outperforms the baseline methods on diverse benchmarks and proves its practical effectiveness as a plug-in module for downstream application tasks.*
>
> > W2: Section 3 is vague about whether the formulation itself constitutes a contribution.
>
> Yes, we are the first to define and formulate this overlooked problem. Previous works, such as the ones listed between lines 63-65, include what we reference as the "naive" method.
>
> To emphasize this point, we suggest the following change to the beginning of section 3:
>
> _`Revision`_: *The scene graph manipulation problem exemplifies the physical rearrangements of scenes. However, naively inserting or changing nodes often breaks commonsense or spatial consistency. A core contribution of our work is to **precisely characterize what constitutes a “reasonable” scene graph** and to **formalize the manipulation operations required for this task**. In this section, we therefore (1) introduce a novel and principled definition of reasonable scene graphs grounded in observed properties, and (2) formally specify the fundamental scene graph manipulation actions — adding, removing, and altering relationships — that serve as the basis of our proposed framework.*
>
> >W3: The method is a straightforward transformer with masked training and does not introduce a particularly novel idea.
>
> We acknowledge this. The training scheme is inspired by next-token pre-training, but the novelty lies in decomposing scene graph manipulation into an iterative cut-and-stitch process suited for next-token learning. Unlike LLM pre-training, where next-token prediction is auxiliary, our model directly solves a real task with it. By doing so, our approach easily adapts to modern transformer architectures; we compare GPT-2-style and LLaMA-3-style models in the experiments section.
>
> >W4.1: MPNN (2019) is selected as the baseline for comparison without justification. It is one of the earliest works in graph neural networks, and many GNNs have since been developed.
>
> Yes, we admit the fact that the MPNN baseline is adapted from an old work (SG-Net). However, the MPNN baseline is still the most relevant work on scene graphs, and it represents networks that adopt the message passing mechanism and are therefore idea-wise different from our method.
>
> Since our method is based on decoder-only transformers and uses architectures proposed for modern large language models, we focus on comparing our methods to finetuning techniques (prompting, parameter-efficient finetuning) of large language models to show that with our novel decomposition of scene graph manipulation, large language model transformers can learn spatial commonsense and solve this task effectively.
>
> >W4.2: Downstream tasks for robotics applications present only qualitative results. Previous results are primarily analytical, and quantitative downstream results are essential to justify the importance of graph manipulation.
>
> We agree quantitative results are important and have included them in Tables 1–3 for both scene graphs and scene generation. This aligns with the flexible 3D data pipeline we envisioned (lines 54–58).
>
> In SG-Bot, success rates depend on precise object positions (e.g., IoU). SG-Tailor focuses on spatial/semantic plausibility. Our cut-and-stitch technique focuses on relation changes but not exact distances, so manipulating multiple edges may deviate more from ground-truth coordinates, affecting SG-Bot metrics. In contrast, Graph-to-3D uses relation-based spatial accuracy (e.g., left-right correctness), which aligns with SG-Tailor, so we include those quantitative results.
>
> >Q1: “Reasonable” itself is somewhat vague. In my understanding, it simply refers to a scene graph that does not conflict with human knowledge
>
>  Please refer to section 3.2 for the definition of reasonable scene graphs. Since a mathematically rigorous definition of "reasonable" is out of the scope of this paper, we define reasonable scene graphs as *"the set of scene graphs that do not violate human intuitions"* and describe some of their properties that come from real-life observations.

---

> ### Author Response · Authors · 2025-11-27
>
> Dear Reviewer AvBN,
>
> We would like to kindly ask for your feedback on our responses to the points you raised, as your input is crucial for the discussion.
>
> Your time and effort are much appreciated.
>
> Best,
> The Authors

---

> > ### Comment · Reviewer_AvBN · 2025-11-28
> >
> > Thank you for your further explanation.
> >
> > From the answers to W1 and W2, a primary contribution lies in proposing the problem definition, while in Q1, it is stated that providing a more rigorous definition is out of scope. In my opinion, if no more rigorous or domain-specific definition is required for scene-graph manipulation, then the contribution of the problem formulation is very marginal, because the listed graph-editing operators are simply basic and commonly used changes in computer science. I cannot find any specific benefit or necessity for sharing this problem formulation.
> >
> > Overall, I acknowledge the contribution of the methodology mentioned in W3, but both the method contribution and the problem-formulation contribution remain weak. Since this point is not about technical flaws but about the overall contribution of the work to the community, I do not expect this opinion to change further. I will keep my score and hope to leave the decision of whether to take this point into account in the final evaluation to the area chair.

---

### Official Review · Reviewer_WnSW · 2025-11-01

**Soundness:** 3
**Presentation:** 4
**Contribution:** 2
**Rating:** 6
**Confidence:** 3

**Summary:**

The authors propose the SG-Tailor framework, which autoregressively reasons inter-object relationships to resolve semantic conflicts during scene graph manipulation. SG-Tailor utilizes a "Cut-And-Stitch" strategy, redefining the graph-level operations into cut and stitch steps and providing a novel perspective. The effectiveness of SG-Tailor is demonstrated through extensive experiments on 3D scene datasets, where it significantly outperforms both MPNN and state-of-the-art LLM baselines in generating coherent graphs.

**Strengths:**

- Clear problem definition and motivation. Well-structured paper which was very easy to follow.
- Comprehensive experiments on various datasets and effective baseline selection on multiple aspects.
- Interesting cut-and-stitch strategy, suggesting a novel viewpoint to graph operations and offering a solid framework.

**Weaknesses:**

- Downstream bottlenecks: As the authors acknowledge, the model's practical utility is constrained by the limitations of the downstream modules (like Graph-to-3D) and the fixed predicate vocabulary of the datasets.
- Novelty seems minor: The paper's contribution seems to lie in its formulation of the scene graph manipulation task, its practical application, and its strong empirical results, rather than on a fundamental architectural or theoretical advance.

**Questions:**

- Some parts of the methodology seem computationally heavy, e.g., brute-force way of editing. Can you provide a comparison on computational complexity with baseline methods? Also, I would like to know the scalability of this method. How does the performance (both speed and accuracy) degrade as the number of objects in a scene graph grows?
- Cycle rates don’t reflect the degree of incoherence (e.g., two or more conflicts) or other forms of spatial contradictions (other than cycles). Can you provide other metrics that can also reflect such aspects?

---

> ### Author Response · Authors · 2025-11-20
>
> Dear Reviewer WnSW,
>
> thanks very much for your review. Your acknowledgment of our problem formulation and method explanation is very important to us.
>
> >W1: The model's practical utility is constrained by the limitations of the downstream modules (like Graph-to-3D) and the fixed predicate vocabulary of the datasets.
>
> Yes, we admit this is a key limitation in the current implementation of SG-Tailor, since the downstream task is not a part of the optimization loop. As we suggest in the limitations part, we see this as the path for future works.
>
> >W2: Novelty seems minor: The paper's contribution seems to lie in its formulation of the scene graph manipulation task, its practical application, and its strong empirical results, rather than on a fundamental architectural or theoretical advance.
>
> Yes, it is true that this work does not focus on the theoretical aspect. As SG-Tailor is based on decoder-only transformers, it can be improved with the future advances in the community.
>
> >Q1: Some parts of the methodology seem computationally heavy, e.g., brute-force way of editing. Can you provide a comparison on computational complexity with baseline methods? Also, I would like to know the scalability of this method. How does the performance (both speed and accuracy) degrade as the number of objects in a scene graph grows?
>
> Thank you for pointing this out.
>
> We want to first discuss that the measurement of the computation overhead of the large language models is not meaningful, as the hosting device specifications are unknown. For the MPNN and our method, they both employ the cut-and-stitch strategy in solving spatial conflicts. We suggest including the computation overhead as a section in the Appendix. Please refer to our response to Reviewer x7WB's Q3.1.
>
> On the scalability of the method, although we did not experiment on this, we purposefully used 3 datasets with varying sizes of scene graphs to test the scalability of our method: 3RScan for small, simple scenes, 3DFront for small but dense scene graphs, and SceneVerse for large-scale scenes. The results can be evaluated across datasets in Table 1: the performance degrades significantly as the scene graphs grow more complex.
>
> >Q2: Cycle rates don’t reflect the degree of incoherence (e.g., two or more conflicts) or other forms of spatial contradictions (other than cycles). Can you provide other metrics that can also reflect such aspects?
>
> We are unsure if other types of spatial contradictions exist in scene graphs. All scene graphs used in the experiment sections generally contain only spatial relationships that can be characterized as cycles, such as left-right, front-back, and above-below (please refer to Table 6 in the appendix). Note that those relationship pairs are not directly coupled - the left cycle can coexist with front cycles. More complex conflicts, such as the bottom left and bottom right can be decomposed into a cycle in the left direction.

---

> > ### Comment · Reviewer_WnSW · 2025-11-28
> >
> > I have read the rebuttal. I am okay with the paper which already reflected in my original rating.

---

### Official Review · Reviewer_x7WB · 2025-11-03

**Soundness:** 3
**Presentation:** 2
**Contribution:** 2
**Rating:** 2
**Confidence:** 4

**Summary:**

This paper presents SG-Tailor, an autoregressive model designed to address conflicts in scene graph manipulation tasks. Scene graphs capture complex relationships among objects and serve as a key component in generating and manipulating 3D scenes. However, existing methods for manipulating scene graphs struggle to handle semantic conflicts that arise when nodes or edges are modified. SG-Tailor introduces the "Cut-and-Stitch" strategy, which allows the model to infer reasonable relationships for newly added nodes and resolve conflicts caused by edge modifications, producing coherent scene graphs. Extensive experiments demonstrate that SG-Tailor outperforms existing methods on multiple benchmarks and can be seamlessly integrated into downstream tasks, such as scene generation and robotic manipulation.

**Strengths:**

1. Importance of the problem selection: Scene graph manipulation is a key challenge in computer vision and robotics, and the paper focuses on solving the issue of relationship conflicts, which has practical application value.

2. Method innovation: The Cut-And-Stitch strategy is a novel approach that decomposes scene graph manipulation into cutting and stitching steps, offering both intuitiveness and effectiveness.

3. Experimental comprehensiveness: The paper evaluates the method across multiple datasets, demonstrating its generalization ability.

4. Practicality and scalability: The method can be integrated as a plug-in module for downstream tasks, such as robotic manipulation.

**Weaknesses:**

1. The core method of the paper is based on autoregressive models, which is a common technique in natural language processing (NLP) and some graph learning tasks. Although applying this to scene graph manipulation is somewhat novel, the paper fails to sufficiently demonstrate the essential differences from existing works, such as SGNet, MPNN, or LLM-based methods.

2. The Cut-And-Stitch strategy lacks theoretical support: The paper claims that the Cut-And-Stitch strategy can effectively resolve conflicts, but it does not provide theoretical analysis or mathematical proof to demonstrate its optimality. For example, how can we ensure that the stitching step always produces a conflict-free graph after the cutting step?

3. Generalization ability is questionable: The paper claims that SG-Tailor can be applied to multi-task learning and downstream applications (e.g., robotic manipulation), but the experimental section only briefly mentions this, lacking detailed results or user study data. For instance, the robotic manipulation experiment (Appendix A) only provides qualitative examples without offering quantitative metrics or comparisons with professional methods.

**Questions:**

1. How does the specific implementation of the Cut-And-Stitch strategy ensure global consistency?
2. How does SG-Tailor handle large-scale scene graphs? When the number of nodes increases, does the sequence length exceed the model's context window?
3.  What is SG-Tailor's performance in real-time systems (such as robotic interactions)? How robust is it to input noise? The paper mentions downstream applications (such as robotic manipulation), but the real-time performance has not been evaluated. If the input scene graph contains annotation errors or noise (such as incorrect relationships), can the model correct them? Figure 8 demonstrates robotic manipulation results, but there is no quantitative analysis of the success rate. Please add task-level metrics (such as planning accuracy, execution efficiency).

---

> ### Author Response · Authors · 2025-11-20
>
> Dear Reviewer x7WB:
>
> Thank you very much for your time. Your reviews bring up some important aspects.
>
> >W1: the paper fails to sufficiently demonstrate the essential differences from existing works, such as SGNet, MPNN, or LLM-based methods
>
> We would like to first draw attention to the fact that SGNet belongs to the MPNN methods. They are not two distinct categories. In fact, we have included a description of each baseline method in section 5.4, and more details on the experiments can be found in the appendix, including the prompts and finetuning protocol.
>
> >W2: The Cut-And-Stitch strategy lacks theoretical support... How can we ensure that the stitching step always produces a conflict-free graph after the cutting step?
>
> Through the optimal substructure property described between lines 170-179, and by learning the knowledge priors in the dataset. Given a graph that faithfully represents a real scene, we cut an object out of the scene, and let SG-tailor stitch the relationships between the object and the rest of the scene. If all relationships inferred by SG-Tailor are correct (and a well-trained model should be able to do that), the resulting graph should be conflict-free.
>
> >W3: ...the robotic manipulation experiment (Appendix A) only provides qualitative examples without offering quantitative metrics or comparisons with professional methods.
>
> Please refer to answer to W4.2 of our response to Reviewer AvBN.
>
> >Q1: How does the specific implementation of the Cut-And-Stitch strategy ensure global consistency?
>
> Please refer to answer to W2.
>
> >Q2: How does SG-Tailor handle large-scale scene graphs? When the number of nodes increases, does the sequence length exceed the model's context window?
>
> Yes, the maximum sequence length is dependent on the context window. GPT2 transformers have a context length of 1024 tokens, and Llama3 transformers have a context length of 8192. This is approximately 1024/8=128 and 8192/8=1024 triplets. 3DFront is a dataset specially curated for dense scene graphs, and its length is generally under 100 triplets. We therefore argue that the current implementation of our method is sufficient for its use.
>
> >Q3.1: What is SG-Tailor's performance in real-time systems (such as robotic interactions)? How robust is it to input noise?
>
> We propose to add a small discussion in the supplementary material. Revision " Computation Overhead. We discuss the computation overhead of our cut-and-stitch method in this section. Since the node of interest and all its associated edges are first removed, then edges to all other nodes are inferred and stitched to the scene graph, we intuitively see a linear increase in the number of model inferences compared to graph density. As discussed in Vaswani et al. (2017), transformer-based architecture scales quadratically in the sequence length. We expect SG-Tailor to also scale quadratically in graph density. The average computation overhead of SG-Tailor is compared with that of MPNN on different datasets, and the results are reported in the table. " We will report the average computation overhead of SG-Tailor in seconds on 3RScan, SceneVerse37k, and 3DFront: 18.81, 57.24, 33.41 on an RTX-2080ti GPU, and 12.71,  27.39, 17.22 for the MPNN baseline.
>
> >Q3.2: If the input scene graph contains annotation errors or noise (such as incorrect relationships), can the model correct them?
>
> No, the current implementation of SG-Tailor can not explicitly detect errors in the input scene graph. We intentionally designed it so, because scene graph manipulation aims to solve conflicts arising from manipulating a real scene that is reasonable.
> However, if one really wants to, an additional error indicator class can be added to the token classes, and SG-Tailor can be retrained on a curated dataset that contains such errors in the input for the detection of incorrect labels.

---

> ### Author Response · Authors · 2025-11-27
>
> Dear Reviewer x7WB,
>
> We would like to kindly ask for your feedback on our responses to the points you raised, as your input is crucial for the discussion.
>
> Your time and effort are much appreciated.
>
> Best,
> The Authors

---

### Official Review · Reviewer_H5rL · 2025-11-04

**Soundness:** 4
**Presentation:** 4
**Contribution:** 3
**Rating:** 6
**Confidence:** 3

**Summary:**

This paper focuses on how to maintain the consistency of the scene graph after inserting or manipulating the node and proposes the SG-Tailor for robust scene graph manipulation.

**Strengths:**

1. The problem statement of scene graph manipulation is valid and convincing.
2. The proposed structure shows significant improvements in the experiment results.

**Weaknesses:**

1. The paper lacks theoretical analysis on the proposed method and the motivation of the SG-Tailor structure is unclear. Why is it a good idea to formulate inter-object relationship reasoning into the autoregressive sequence generation task? It will be good if authors can provide any bounds based on equation 7.
2. The result on SceneVerse37K is only compared with MPNN. A single baseline is not sufficient to claim the effectiveness of the proposed structure. Can authors give the results from other models?

**Questions:**

1. In table 4, why there are two MPNN appears in the 3D-FRONT dataset section. Are they different?
2. For the cycle rates, table 2 shows great improvement to avoid generating unnecessary parts of the graph. What is the cost for that? Will SG-Tailor take longer to generate the graph parts? In the use case mentioned in appendix A, what is the overall execution time for the baseline and the SG-Tailor?

---

> ### Author Response · Authors · 2025-11-20
>
> Dear reviewer H5rL:
>
> Thank you very much for your time, and thanks for acknowledging our paper's problem formulation and results.
>
> >W1: Why is it a good idea to formulate inter-object relationship reasoning into the autoregressive sequence generation task? It will be good if authors can provide any bounds based on equation 7
>
> This is a good question. The motivation of the autoregressive formulation comes from the intuition that people would populate a real indoor scene with objects one by one. The placement of later objects is highly dependent on the previous objects. Autoregressive modeling is a very natural abstraction of this. As to the mathematical foundation of next-token learning, we argue that this is beyond the scope of this work, and it is a well-studied topic discussed by many great theory papers, such as *A Neural Probabilistic Language Model*, Bengio et al., 2003.
>
> >W2: The result on SceneVerse37K is only compared with MPNN. A single baseline is not sufficient to claim the effectiveness of the proposed structure. Can authors give the results from other models?
>
> Due to the practical budget limit, unfortunately, we can only afford to conduct experiments of large language models on a smaller subset, SceneVerse100, to show the effectiveness of our method on this type of scene graphs. The results are summarized in Appendix D, Table 4. We still show the results on SceneVerse73k to show that our method is scalable compared to the MPNN baseline.
>
> >Q1: In table 4, why there are two MPNN appears in the 3D-FRONT dataset section. Are they different?
>
> Thank you so much for pointing out our mistake! We compared the manuscript with the historical version and found that the first line of the MPNN result is actually the result on SceneVerse100. Table 4 is an extension of Table 1, where large language model baselines are also included for the datasets 3RScan and SceneVerse100. No new entries should be included for the dataset 3DFront. We propose to move this row up under SceneVerse100 our final version.
>
> >Q2: For the cycle rates, table 2 shows great improvement to avoid generating unnecessary parts of the graph. What is the cost for that? Will SG-Tailor take longer to generate the graph parts? In the use case mentioned in appendix A, what is the overall execution time for the baseline and the SG-Tailor?
>
> Yes, the cost of adopting our method is computation overhead. We have proposed to add a specific section in the appendix to report this. Please refer to our response to Reviewer x7WB's Q3.1.

---

### Meta-Review · Area_Chair_yn2N · 2026-01-07

**Summary:**

The reviewers raised concerns about limited novelty, insufficient experimental evaluation, and lack of theoretical support/analysis. The main concerns—specifically, insufficient experiments and generalizability—remain unresolved. Therefore, I recommend rejecting this paper.

**Reviewer Concerns:**

- Reviewer H5rL: lack of theoretical analysis, insufficient baseline methods, lack of runtime analysis
- Reviewer x7WB: limited novelty, lack of theoretical support, limited generalizability
- Reviewer WnSW: Downstream bottlenecks, limited novelty, computational complexity
- Reviewer AvBN: limited novelty, insufficient experiments

**Reviewer Scores:**

- Reviewer H5rL: No
- Reviewer x7WB: No
- Reviewer WnSW: No
- Reviewer AvBN: No

---

### Decision · Program_Chairs · 2026-01-26

Reject